# Exercise Preferences and Benefits in Patients Hospitalized with COVID-19

**DOI:** 10.3390/jpm12040645

**Published:** 2022-04-17

**Authors:** Sevasti Kontopoulou, Zoe Daniil, Konstantinos I. Gourgoulianis, Ourania S. Kotsiou

**Affiliations:** 1Department of Respiratory Medicine, University of Thessaly, 41110 Larissa, Greece; sevi_kon@hotmail.com (S.K.); zdaniil@uth.gr (Z.D.); kgourg@med.uth.gr (K.I.G.); 2Faculty of Nursing, University of Thessaly, 41110 Larissa, Greece

**Keywords:** dyspnea, exercise, hospitalization, recovery

## Abstract

Background: Obese people are at risk of becoming severely ill due to SARS-CoV-2. The exercise benefits on health have been emphasized. Aim: To investigate the correlation of obesity with the length of hospitalization, the pre- and post-hospitalization exercise preferences of COVID-19 patients, and the impact of pre-admission or post-hospitalization physical activity on dyspnea one month after hospitalization and recovery time. Methods: A telephone survey was conducted in patients hospitalized at the Respiratory Medicine Department, University of Thessaly, Greece, from November to December 2020. Results: Two-thirds of the patients were obese. Obesity was not associated with the hospitalization time. Two-thirds of the patients used to engage in physical activity before hospitalization. Males exercised in a higher percentage and more frequently than women before and after hospitalization. The methodical pre-hospitalization exercise was associated with lower levels of dyspnea one month after hospitalization. In-hospital weight loss, comorbidities, and dyspnea on admission independently predicted longer recovery time. Lockdown had boosted men’s desire to exercise than females who were negatively affected. Conclusions: Obesity is common in COVID-19 hospitalized patients. In-hospital weight loss, comorbidities, and dyspnea on admission predicted a longer post-hospitalization recovery time. The pre-hospitalization exercise was associated with less post-hospitalization dyspnea and recovery time.

## 1. Introduction

COVID-19 is a multisystemic and multivessel disease that involves the respiratory, cardiovascular, renal, gastrointestinal, and central nervous systems. The presence of comorbidities increases the risk for severe illness due to SARS-CoV-2. More often, people with underlying medical conditions display respiratory failure that requires admission to the ICU, multiorgan failure, or even loss of their lives [1].

Obesity exposes infected individuals to peril, increasing the required days of hospitalization and recovery. This connection occurs because the chronic storage of body fat is directly associated with a chronic pro-inflammatory state, weakening the immune system and creating an ideal environment for the virus to grow inside the fat cells. Obesity, combined with comorbidities, aggravates the symptoms of the COVID-19 disease by extending the time of needed hospitalization and eventually raising the mortality rate. Regular body fat storage and a sedentary lifestyle adopted due to the mandatory quarantine can create the perfect conditions for infection and growth of contagious pathogens, such as the SARS-CoV-2 virus among the more vulnerable population [2].

Consequently, a risk factor for severe illness and needed admission to the Intensive Care Unit (ICU) is the increased body mass index (BMI), namely the increased storage of body fat [3]. A recently conducted study pointed out that the need for intensive mechanical ventilation for COVID-19 patients under 60 was seven times higher for those with a BMI over 35 kg/m^2^ compared to those with a BMI under or equal to 25 kg/m^2^ [4].

Starting on 4 May 2020, a 42-day strict lockdown was implemented in Greece. Movements of individuals to serve their needs outside the house were permitted only for seven categories of reasons: (i) transition to the workplace during work hours; (ii) going to the pharmacy or visiting a doctor; (iii) going to a food store; (iv) going to the bank for services not possible online; (v) helping a person; (vi) going to a significant ritual (funeral, marriage, baptism) or movement, for divorced parents, which was essential for contact with their children; and (vii) moving outdoors for exercising or taking one’s pet out, individually or in pairs. Again, from 7 November 2020, Greece implemented new measures and restrictions on movement and business activity. Kindergartens, primary and special schools initially remained open, and from 18 November 2020, they switched to distance learning. On 14 December 2020, shops, hairdressers, and other facilities were allowed to open, while schools and restaurants remained closed [5].

At the beginning of the quarantine, people generally adopted a sedentary lifestyle with decreased physical activity, ultimately harming their physical and psychological health and their quality and quantity of sleep [6]. Multiple vulnerabilities and an interplay leading from simple anxiety to clinical depression and suicidality through distress were revealed among the Greek population [7].

Moderate to intense exercise entrains important positive adjustments to the cardiorespiratory ability, reduces the levels of chronic inflammation that may have preexisted due to related diseases, and improves the immune system’s function for a faster reaction against viral infection such as COVID-19 [8], improves lipid profile, reduces the BMI, and can have a positive effect on our psychological health [9]. Moreover, physical exercise can help individuals maintain their muscle mass, good respiratory function, and boost the immune system to keep respiratory function levels high [10].

An international online survey including 41 research institutions from Europe, Western-Asia, North-Africa, and the Americas, documented that COVID-19 lockdown deleteriously affected physical activity and sleep patterns [11]. A recently conducted Greek study supported that during a pandemic, compared with a typical week, physical activity of a high and moderate intensity decreased for 43.0% and 37.0% of participants, did not change in 32.9% and 36.1% of participants, and increased only in 24.1% and 26.9%, respectively, whereas walking time decreased in 31.3%, did not change in 27.3%, and increased in 41.5% of participants [12]. In fact, after consecutive lockdown periods were examined, a decline in overall physical activity was evident in all age and gender groups during each lockdown phase [13].

During the pandemic, health regulators constantly point out, mainly to the elderly confined by the quarantine, to avoid a sedentary way of life and engage in any form of physical activity [8]. Furthermore, recent studies mention that exercise reduces hospitalizations due to the COVID-19 disease [14].

This study aimed to investigate the correlation of obesity with the duration of hospitalization, the pre–and post-hospitalization exercise preferences of COVID-19 patients, and the impact of pre-admission or post-hospitalization physical activity on dyspnea one month after hospitalization.

## 2. Materials and Methods

### 2.1. Procedure

This retrospective study was conducted via a telephone survey from February 2021 to March 2021. One specialized trainer was the researcher of this study and contacted previously hospitalized patients via telephone to interview by asking them a list of predetermined questions.

### 2.2. Participants

In the study were included all patients regardless of their age who were infected by the SARS-CoV-2 virus and were hospitalized at the Infection Diseases Unit (COVID-19) of the Department of Respiratory Medicine of the University of Thessaly from November to December 2020. An exclusion criterion was the inability to acquire information due to their non-consent or lack of good cooperation.

### 2.3. Study Tools

In this study, we used an original self-reported questionnaire with 26 questions regarding:(i)Demographics and the body metrics data of every patient before and after hospitalization (age, weight, height, body mass index were recorded);(ii)The presence and types of comorbidities;(iii)The dyspnea levels according to the Modified Medical Research Council (mMRC) dyspnea scale before and one month after hospitalization.Modified Medical Research Council (mMRC) Dyspnea Scale.Dyspnea is defined as the feeling of difficult and labored breathing that results from insufficient aeration. The mMRC scale used by the Medical Research Council classifies dyspnea from level 0 (dyspnea only during intense exercise) to level 4 (severe dyspnea that prevents individuals from leaving their home or even getting dressed) [8].(iv)The length of hospitalization, (v) the physical state before and after hospitalization. As parameters of physical activity in our study, resistance training, either using the bodyweight (Pilates, yoga) or with the assistance of additional equipment (weights, resistance equipment, TRX), aerobics (running, walking, cycling, swimming), as well as work-related physical activity were included, (vi) the number of days per week spent on exercising before and after hospitalization, (vii) the different kinds of exercise before and after hospitalization, and (viii) whether the enforcement of restrictive measures negatively affected the frequency or the desire to exercise. 

### 2.4. Statistical Analysis

The statistical analysis was conducted with the IBM SPSS v23. The quantitative variables were presented as mean value ± standard deviation (SD), and the qualitative variables were presented as an absolute value (frequency). The frequencies were compared with the chi-square statistical test. The *t*-test was used to test the difference between two mean values from independent samples. The nonparametric data were analyzed with the Mann–Whitney U test. The parametric data that compare three or more groups were analyzed with the ANOVA unidirectional Variance Analysis and the post hoc Bonferroni multiple comparison test. In contrast, the nonparametric data were analyzed with the Kruskal–Wallis test and the Dunn multiple comparison test. Spearman’s correlation was used for the correlation analysis. A multiple linear regression model was utilized to examine a series of prediction variables to find those that can better predict faster recovery time in days.

## 3. Results

### 3.1. Demographics, Clinical Parameters, and Symptomatology of the Hospitalized COVID-19 Population

In total, 42 men (66%) and 22 women (34%) were included in the study, with a mean age of 62.2 ± 13.2 years old (min = 21 years, max = 91 years). The demographic and clinical parameters of the study’s population and comparisons according to sex are presented in Table 1.

The men were significantly taller and heavier than the women, as expected (Table 1). The study’s patients were overweight on average with no BMI difference between the genders. A total of 45.3% of the patients (29 patients) were overweight, while 23.4% (15 patients) were obese. Only 31% of the patients had normal weight. The comorbidities of the sample hospitalized due to COVID-19 and their comparison based on sex can be found in Table 2.

The most common comorbidity of the sample was hypertension. The women suffered from DM in a greater frequency than men. The patients with comorbidities were significantly older than those without (65 ± 12 vs. 57 ± 13 years old, *p* = 0.019), as expected.

Older age was positively associated with the level of mMRC dyspnea on admission (r = 0.402, *p* = 0.001) and after hospitalization (r = 0.280, *p* = 0.025). The number of comorbidities was positively associated with the level of mMRC dyspnea before (r = 0.499, *p* = 0.001) and after (r = 0.298, *p* = 0.050) hospitalization. Patients hospitalized due to COVID-19 with high blood pressure, or another cardiovascular disease displayed higher levels of mMRC dyspnea at the time before their admission to the hospital by comparison with those who did not have any comorbidities (2 ± 1 vs. 1 ± 1, *p* < 0.001). Patients with COPD hospitalized due to COVID-19 also displayed higher levels of mMRC dyspnea before their hospital admission than those who had no comorbidities (4 ± 1 vs. 1 ± 1, *p* < 0.001).

The symptomatology of the study’s population on admission and the comparisons based on the presence or absence of comorbidities are presented in Table 3.

The most frequently displayed symptom during hospital admissions among the study’s population was fever. There was no differentiation on the symptoms during admission between genders. Patients with at least one comorbidity more often had dyspnea and fever than the healthy patients prior to infection (Table 3). There was no significant difference in length of stay between patients with and without comorbidities.

### 3.2. Physical Activity Preferences of the Population before Hospitalization

The physical activity of the study’s population before hospitalization and their comparison based on sex can be found in Table 4.

In total, 65% of the sample mentioned their engagement with at least one form of physical activity before being hospitalized. The men were exercising at a higher percentage than the women before their hospitalization and with greater frequency (Table 4). Overall, 51.6% of the study’s population engaged in physical activity mentions walking as their primary exercise before hospitalization despite gender (Table 4).

The patients with at least one chronic disease were used to engage less in aerobic exercise (!8.2% vs. 2.4%, *p* = 0.044) and resistance training (18.2% vs. 0%, *p* = 0.010) compared to individuals without comorbidities.

### 3.3. Physical Activity Preferences of the Population after Hospitalization

The post-hospitalization physical activity of the study’s population and their comparison based on sex is presented in Table 5.

Overall, 68.8% of the study’s population started or continued to exercise after their hospitalization. Men continued to exercise on a larger scale and at a greater frequency than women after their hospitalization (Table 5). In 57.8% of the abovementioned population engaged in any form of physical activity after their hospitalization mentioned walking as their primary physical activity, with the male percentages being significantly higher compared to those of females (Table 5). In 36 out of the 42 individuals (85.7%) exercising before their hospitalization continued to exercise after it. In 8 out of the 44 individuals (18.2%) exercising after their hospitalization had just started, and they were not before.

### 3.4. Changes of Body Weight before and after Hospitalization, and Views about the Effects of Hospitalization or Lockdown on the Frequency and the Desire of Exercise after Discharge

The change of body weight before and after hospitalization, the various views about the effects of hospitalization or lockdown on the frequency and the desire of exercise after discharge and comparisons based on sex are shown on Table 6.

The BMI was not associated with the duration of hospitalization. However, the days of hospitalization were positively associated with more extensive changes in the patients’ weight (r = 0.809, *p* < 0.001). Patients who experienced in-hospital weight loss were hospitalized more days than those who had gained weight (19 ± 14 vs. 9 ± 4, *p* < 0.001).

At a significantly higher percentage than the women, the men stated that their hospitalization or lockdown measures did not affect their exercise frequency after discharge from the hospital. In total, 50% of the men (a more considerable percentage than the women at 9%) supported that the restrictive measures did not affect the frequency of their desire to exercise. Actually, among men, 38.1% that were not previously exercising mentioned that the lockdown increased their desire to begin. Approximately half of women (45.5%) were negatively affected by the lockdown regarding their frequency and desire to exercise, a significantly higher percentage compared to the men.

Both the men and the women that did not exercise (before or after hospitalization) were those individuals who supported the harmful effects of the lockdown to their desire for exercise in comparison to those physically active before their hospitalization. The men and the women that believed in the positive effects of their hospitalization to their desire for exercise were also those who supported the positive effects of the restrictive measures to that desire.

The frequency of exercise before hospitalization was positively associated with the frequency of exercise after hospitalization (r = 0.645, *p* < 0.001). The patients previously engaged in physical activity needed significantly lesser time to recover (22 ± 14 vs. 65 ± 32 days, *p* < 0.001) and displayed significantly lower levels of dyspnea on the mMRC scale (1 ± 1 vs. 3 ± 1, *p* < 0.001) after their hospitalization compared to the patients with no history of physical activity. In fact, the frequency of exercise (days per week) was negatively associated with the levels of mMRC dyspnea after hospitalization (r = −0.342, *p* = 0.026). On the contrary, the recovery time in days was positively associated with the time of hospitalization (r = 0.408, *p* = 0.001). Of all forms of exercise, walking was associated with a shorter recovery time (23 ± 15 vs. 55 ± 34, *p* < 0.001) and a lower score on the mMRC dyspnea scale (1 ± 1 vs. 2 ± 1, *p* = 0.04) in comparison to the absence of any previous physical activity.

The patients engaged in physical activity after their hospitalization displayed significantly lower levels of dyspnea on the mMRC scale (1 ± 1 vs. 3 ± 1, *p* = 0.001) compared to those who did not engage in any physical activity after their hospitalization. Of all the different kinds of post-hospitalization exercise, walking was positively associated with the most remarkable improvement of dyspnea after hospitalization (1 ± 1 vs. 2 ± 1, *p* = 0.004).

A multiple linear regression model was used to research those parameters that can independently predict a quicker recovery. The weight loss in Kg, the presence of chronic disease, and dyspnea on admission, were found to be independent predictors of a faster recovery time (in days) of patients hospitalized due to COVID-19 (R^2^ = 92.0, adjusted R^2^: 89.4) (Table 7).

## 4. Discussion

In this study, we investigated the correlation of obesity with the duration of hospitalization, the pre- and post-hospitalization exercise preferences of COVID-19 patients, and the impact of pre-admission or post-hospitalization physical activity on dyspnea one month after hospitalization or recovery time after discharge. We found that two-thirds of the patients were obese; however, obesity was not associated with the hospitalization time. Two-thirds of the patients used to engage in physical activity before hospitalization. The men exercised in a higher percentage and more frequently than women before and after hospitalization. The methodical pre-hospitalization exercise was associated with lower levels of dyspnea one month after hospitalization and less recovery time in days. Weight loss in Kg, preexisting comorbidity, and dyspnea on admission were independent predictors for faster recovery time in days. Most males used to exercise before infection supported that the lockdown had boosted their desire to exercise, compared to females who were negatively affected.

In our study, the average age of the sample of hospitalized patients in November and December 2020 was 62.2 ± 13.2 years old. It is distinctively mentioned that the frequency of infection is increased for men over 60 years old, who also display higher mortality rates than the women of the study up to 50%. According to the World Health Organization, there were expected differences between the two genders regarding the body metric data, while the population studied was overweight on average, without any statistically significant difference among genders.

Comorbidities were positively associated with age, as expected. It has been reported that 60 to 90% of patients who need hospitalization due to COVID-19, display at least one comorbidity [15,16]. The most common comorbidities mentioned in the literature are hypertension (57% of the patients), respiratory diseases (10% of the patients), and malignancy (8% of the patients) [15,16]. In a recent meta-analysis of 10 studies that included 76,993 patients, the prevalence of high blood pressure was 17.37% (95%CI, 10.15–23.65%), of cardiovascular disease 12.11% (95%CI 4.40–22.75%) and of DM 7.87% (95%CI 6.57–9.28%) [17]. Accordingly, the present study revealed that two-thirds of the population had at least one of the following comorbidities in descending order, high blood pressure (29.7%), DM (21.9%), cardiovascular diseases (CD, CVA) (18.7%), hypercholesterolemia/dyslipidemia (15.6%), malignancy (9.3%), coexistent autoimmune disease (4.7%), COPD (4.7%), asthma (3.1%), chronic hepatitis (3.1%), and thyroid disease (3.1%).

It has been reported that patients with more than one comorbidity experience more dyspnea [18]. In the present study, there was a positive association of dyspnea before and after hospitalization with the presence of comorbidities. Furthermore, advanced age was positively associated with the level of mMRC dyspnea on admission and after discharge. It has been reported that at least 87% of the infected by the virus still displayed at least one of the common symptoms after their recovery, more often dyspnea and fatigue, while 15% of the examined patients displayed increased breathing difficulty as a complication of the virus [19].

Patients with increased BMI are more likely to be severely infected by the new coronavirus. A large percentage of patients that need intensive care are overweight or obese [20]. We found that increased BMI was not associated with hospitalization time. However, it is essential to point out that the patients with more weight included in this study were exercising more often before their hospitalization, and the benefits of their exercise could possibly counterbalance their increased BMI. Physical activity is an important means of promoting health. In addition to improving functions related to cardiovascular and respiratory function, as well as avoiding the deposition of body fat, physical activity has many benefits for avoiding infectious diseases, or in the case of hospitalization, for faster recovery [10,21].

More specifically, physical activity improves inflammation associated with COVID-19 independent of body fat, explaining why bodyweight was not an independent contributor to hospitalization. There are data demonstrating that regular bouts of short-lasting (i.e., 45–60 min), moderate-intensity exercise (50–70% VO2max), performed at least three times per week is beneficial for the host immune defense, particularly in older adults and people with comorbidities [22,23], compared to prolonged and/or intense bout of endurance exercise that makes humans more susceptible to infection. Moderate-intensity exercise has been linked to increased leukocyte function in humans [24]. It has been found to enhance chemotaxis, degranulation, phagocytosis, cytotoxic activity, and the oxidative activity of macrophages and neutrophils in rats [25]. Increased cytolytic activity of NK cells and NK cell-activating lymphokine during 60 min of moderate-intensity exercise by healthy cyclists was also reported [11]. On the contrary, the long-duration/intense exercise-induced immunomodulation is associated with markers of immunosuppression, such as increased production of proinflammatory cytokines [26] reduced activity of NK cells, T and B lymphocytes, and neutrophils; reduced production of salivary IgA and plasma IgM and IgG; and a low expression of major histocompatibility complex II in macrophages [27,28]. These changes can be detected hours to days after the end of a prolonged and/or intense endurance exercise. Physical activity controls the viral gateway, modulates inflammation, stimulates NO production pathways, and establishes control over oxidative stress. Adaptation to usual exercise appears to affect immune function, particularly innate and adaptive immunity, and improve humoral immunity with increased vaccination responses. Exercise may at least partially counteract the detrimental effect of SARS-CoV-2 binding to the angiotensin-converting enzyme receptor [22]. Physical can activate anti-inflammatory signaling pathways. In this regard, the release of anti-inflammatory cytokines from skeletal muscle contraction, cortisol elevations, prostaglandin E2, and soluble receptors against tumor necrosis factor and interleukin 2, and increased mobilization of immunoregulatory leukocyte subtypes may be relevant in attenuating the cytokine release in COVID-19. Exercise may enhance alternative routes of NO production, stimulating eNO with antiviral effects and post-infection lung recovery of COVID-19. The control of oxidative stress, which produce cell damage, is modulated by the practice of physical activity by two mechanisms, the inhibition of NF-κB, and the stimulation of Nrf2 pathways [22].

A total of 65% of the studied sample mentioned that they engaged at least in one form of physical activity before their hospitalization, with walking being the most common. The men, compared to the women, were exercising more frequently both before and after their hospitalization. Our results are in agreement with previous studies documenting that woman were less active than men [29] and that levels of physical activity decreased progressively with age especially among women [30]. Another study reported that women were less likely than men to prefer activities that require skill and practice or done outdoors [31].

Conversely, a recent Italian study reported that women, who previously had a lower level of physical activity than men, showed a lower tendency to reduce it during lockdown, revealing greater resilience than men. During that period, women were motivated by weight loss and toning more than men [32,33], being concerned with controlling their weight, improving their physical appearance, or counteracting the effects of aging. In the present study, the factor that affected their pursuit of physical activity differs between the two sexes, and the leading cause of this phenomenon may be that Greek women tend to spend more time handling family matters and everyday family needs [34], despite the fact that females reported higher motivation for appearance and physical condition than males [33]. A study that was conducted before the COVID-19 pandemic documented that half of the studied population were physically inactive, indicating that sedentary lifestyles have become a serious epidemic in Greece [35].

An interesting finding of this study was that the patients methodically exercising had lower levels of dyspnea one month after hospitalization, needed less time to recover from the infection and return to their previous physical condition. Out of all kinds of exercise we included, walking was positively associated with a faster recovery time of the hospitalized patients. Given that in the present study the age group is over 60, we assume that it is not the walking itself in regard to intensity and how it affects the body but the fact that we had a high number of individuals in that category. Walking is a type of physical activity that is relevant to older adults as the walking ability is of primary importance for older adults [36]. On the other hand, there are data demonstrating that regular bouts of moderate-intensity exercise performed at least three times per week is more beneficial in immunomodulation compared to high intensity exercise, as mentioned above.

Exercise positively affects the immune system contributing considerably to its improvement, bearing in mind the kind, the intensity, and the duration of exercise [37]. Overall, it is a fact that mild exercises stimulate cellular immunity, increase the anti-infective activity of the macrophages and the effect of the inflammatory cytokines, contributing to the faster cure from the infection [37]. There are no data regarding the improvement of the immune response to COVID-19 infection through exercise; however, there are indications, from past viral infections of the respiratory system, of physical activity decreasing the duration and the severity of the symptoms, as well as the mortality rates of the viral disease. Physical activity of mild intensity could be considered nonmedicinal means to the fight against respiratory infections [37].

Regarding the physical activity after hospitalization, walking was again the main preferable exercise by 57.8%. It is essential to mention that this percent value increased (up to 68.8%) because most patients continued to exercise, and at the same time, some patients had begun to exercise after their recovery. Similarly, in that situation, there were more men who continued to exercise than women. In addition, the men stated that the hospitalization and restrictive measures did not affect their frequency or desire to exercise and they continued to work out the same after being cured, some even more, however, the women were affected negatively and reduced their exercise frequency.

A percentage of men and women believed that their hospitalization and the restrictive measures increased their desire to exercise. Mainly, the patients who exercised before infection claimed that their desire decreased, and their exercise after hospitalization became even more intense. On the other hand, the patients that were not exercising at all continued to keep their distance from physical activity and demonstrated that the confinement and the hospitalization affected their desire to exercise negatively. A large proportion of patients (20%) with COVID-19 will continue to have clinical manifestations of the disease, such as fatigue, weakness, severe dyspnea, and headaches for a period that may exceed one month. There is considerable evidence that physical activity has long-term health benefits that mitigate or even prevent the development of chronic non-communicable diseases (lung disease, heart disease, neurocognitive problems, musculoskeletal problems). On the other hand, physical inactivity has been associated with serious COVID-19 problems, including dyspnea [38]. Accordingly, the Centers for Disease Control and Prevention (CDC) advise not only to engage the inactive population in physical activity, but also to establish it as a tool in the management of patients with post-COVID-19 syndrome. Since exercise has been shown to be beneficial for many viral infections such as COVID-19, it is worth highlighting and further examining the extent of the favorable impacts of exercise [38]. It is also important to mention that moderate physical activity significantly increases the anti-pathogenic activity of macrophages, increases the circulation of immune cells, immunoglobulins, and anti-inflammatory cytokines while reducing the possibility of organ damage (such as the lung) due to COVID-induced inflammation [38]. Therefore, physical exercise is shown to be a non-pharmacological intervention that achieves immune enhancement and reduces the negative effects of the disease [36].

Nevertheless, it is crucial to direct our attention to that significant part of the examined sample (36.4%) that did not exercise before the lockdown and started to, not so much, of course, as those who were already used to exercise regularly. A lower frequency of exercise is imperative to eventually adopt a healthier way of life in general than no physical activity at all. This mainly happened due to the shutdown of almost all businesses which left more free time for people. At the same time, while physical activity is the only way of transportation outside the house, even for an hour, a motive has been provided to a vast part of the population to engage in physical activities and follow a healthier way of living [38].

The recovery time was positively associated with the days of hospitalization, as the more days a patient was hospitalized, the more time they needed to return to their physical condition before infection. The data of many studies support that those patients hospitalized for a long time or subjected to invasive ventilation for a prolonged period displayed respiratory and muscle difficulties, a key factor for their recovery time and the restoration of their previous physical condition to how it was before hospitalization [39].

Together, in the present study we found that preexisting comorbidities, dyspnea on admission, and weight loss in Kg during hospitalization were independent predictors for a longer recovery time after the hospitalization. The days of hospitalization were associated with more significant weight fluctuations, either weight gain or weight loss. In total, 56.3% of the patients displayed weight loss. This study showed that weight loss in Kg is an independent indicating factor for greater needed time for recovery. The severe inflammation caused by the virus and results in the release of much more acute phase proteins disorganizes the metabolism and causes weight loss. Additionally, the decreased food intake mostly connected with loss of appetite due to the disease’s symptoms is one more main factor. Another important cause of weight loss is the anxiety brought on the surface due to the disease and bad sleep quality during hospitalization. In addition, immobility due to hospitalization undoubtedly contributes to muscle atrophy, a decrease in adipose tissue, sarcopenia, and, eventually, weight loss. The decreased weight and cachexia due to hospitalization increase the time patients need to return to their state prior to infection [39].

Several limitations need to be noted regarding the present study. A main limitation is that the data-stream provided by self-reporting is not shielded from potential acquiescence response bias. For this reason, the self-reported administered questionnaires were used, measured physical activities that were relevant to older adults over a relatively short period of time before and after hospitalization to minimize reporting errors [40]. In addition, reliability measures were not used in the study, but only self-report questionnaires were used to collect data. Moreover, the physical activity is not the only predictor for obesity but there are other factors as well that should be evaluated in future studies. Furthermore, we do recognize that our study was obviously limited by the small sample size. Even though we aimed to have a larger sample size, the actual response rate was much lower. Nevertheless, this study for the first time evaluated the frequency and type of physical activity among adults previously hospitalized due to COVID-19 in Greece.

## 5. Conclusions

We found that two-thirds of the hospitalized patients were overweight or obese. The increased BMI was not associated with the hospitalization time. This study also showed that weight loss in Kg, a pre-existing chronic disease, and dyspnea as a symptom during hospital admission could independently predict a longer recovery time. Two-thirds of the patients used to engage in some form of physical activity before infection. The men were exercising in a higher percentage and more frequently than the women before their hospitalization. Most of the patients that used to exercise before infection supported that the lockdown had boosted their desire to exercise. At a significantly higher percentage than women, men supported that their hospitalization and the restrictive measures did not affect their frequency or desire to exercise after discharge from the hospital. Women in their majority were negatively affected by the lockdown, at a higher percentage than men, regarding their frequency and desire to exercise.

Hence, obesity is a common comorbidity in patients with COVID-19 that was not proven to be associated with recovery time. Physical activity has long-term health benefits in COVID-19 patients given that those with methodical contact with exercise before infection had low levels of dyspnea after their hospitalization and less recovery time. Avoiding a sedentary life and adopting a healthier way of living by engaging in any form of physical activity are proven to positively affect the rehabilitation from the immensely severe COVID-19 disease that requires hospitalization.

## Figures and Tables

**Table 1 jpm-12-00645-t001:** Demographic and clinical data of the study’s population and their comparison according to sex.

Parameters	Sum n = 64	Men n = 42	Women n = 22	*p*-Value
Age	62.2 ± 13.2	61.1 ± 11.9	64.2 ± 15.5	0.385 *
Height (cm)	171.0 ± 10.3	176.2 ± 7.5	161.0 ± 7.1	<0.001 *
Weight	81.6 ± 14.2	84.9 ± 13.2	75.2 ± 13.4	0.010 *
BMI	28 ± 4	27 ± 3	29 ± 5	0.113 *
mMRC before hospitalization	1 ± 1	1 ± 1	1 ± 1	0.724 *
mMRC after hospitalization	2 ± 1	2 ± 1	2 ± 1	0.068 *

Note: The data are presented as mean value ± SD; * Student’s *t*-test.

**Table 2 jpm-12-00645-t002:** Comorbidities of the study’s population and their comparison based on sex, n = 64.

Parameters	Sum n = 64	Men n = 42	Women n = 22	*p*-Value
At least one comorbidity	42 (65.5)	26 (61.9)	16 (72.7)	0.280 *
Number of comorbidities	2 ± 1	2 ± 1	2 ± 1	0.258 *
Hypertension	19 (29.7)	10 (23.8)	9 (40.9)	0.129 *
Cardiovascular diseases (CD, CVA)	12 (18.7)	7 (16.7)	5 (22.7)	0.392 *
DM	14 (21.9)	6 (14.3)	8 (36.4)	0.046 *
HDL	10 (15.6)	5 (11.9)	5 (22.7)	0.218 *
Malignancy	6 (9.3)	3 (7.1)	3 (13.6)	0.336 *
Autoimmune diseases	3 (4.7)	1 (2.4)	2 (9.1)	0.270 *
COPD	3 (4.7)	3 (7.1)	0	0.270 *
Asthma	2 (3.1)	1 (2.4)	1 (4.5)	0.573 *
Chronic Hepatitis	2 (3.1)	0	2 (9.1)	0.427 *
Thyroid Disease	2 (3.1)	2 (4.8)	0	0.427 *

Note: The data are presented as frequencies (percentages) or mean values ± SD; * Chi-square Test of independence; Abbreviations: CD, coronary disease; CVA, cardiovascular accidents; DM, diabetes mellitus; COPD, chronic obstructive pulmonary disease; HLD, hyperlipidaemia.

**Table 3 jpm-12-00645-t003:** Population’s symptomatology during admission and their comparison based on the presence or absence of comorbidities n = 64.

Symptomatology	Sum n = 64	Presence of Comorbidities (n = 42)	Absence of Comorbidities (n = 22)	*p*-Value
Number of symptoms	2 ± 1	2 ± 1	2 ± 1	0.648
Fever	54 (84%)	32 (76.2%)	22 (100%)	0.010 *
Dyspnea	29 (45%)	24 (57.1%)	5 (22.7%)	0.008 *
Cough	22 (34%)	14 (33.3%)	8 (36.4%)	0.510
Sore throat	7 (11%)	5 (11.9%)	2 (9.1%)	0.546 *
Abdominal pain	4 (6%)	3 (7.1%)	1 (4.5%)	0.574 *
Joint pain	4 (6%)	1 (2.3%)	3 (13.6%)	0.113 *
Myalgia	4 (6%)	3 (7.1%)	1 (4.5%)	0.574 *
Loss of taste	3 (5%)	1 (2.4%)	2 (9.1%)	0.730 *
Expectoration	2 (3%)	1 (2.4%)	1 (4.5%)	0.427 *
Loss of smell	0	0	0	-
Nasal discharge	0	0	0	-
Days of hospitalization	15 ± 12	16 ± 13	12 ± 7	0.129 *

Note: The data are presented as frequencies (percentages) or on average ± SD; * Chi-square Test of independence.

**Table 4 jpm-12-00645-t004:** Physical activity of the study’s population before hospitalization and comparisons based on sex, n = 64.

Parameters	Sum n = 64	Men n = 42	Women n = 22	*p*-Value
Physical exercise before hospitalization	42 (65%)	32 (76.2%)	10 (45.5%)	0.015 *
Days of exercising per week	5 ± 1	5 ± 1	4 ± 1	0.023
Walking	33 (51.6%)	24 (57.1%)	9 (40.9%)	0.166 *
Activity at work	18 (28.1%)	13 (31.1%)	5 (22.7%)	0.348 *
Resistance training	4 (6.2%)	4 (9.5%)	0	0.176 *
Aerobic exercise	5 (7.8%)	5 (11.9%)	0	0.112 *
TRX	2 (3.1%)	2 (4.8%)	0	0.427 *
Cycling	0	0	0	-

Note: The data are presented as frequencies (percentages) or mean values ± SD; * Chi-square Test of independence.

**Table 5 jpm-12-00645-t005:** Post-hospitalization physical activity of the study’s population and their comparison based on sex n = 64.

Parameters	Sum n = 64	Men n = 42	Women n = 22	*p*-Value
Physical activity	44 (68.8%)	34 (81%)	10 (45.5%)	0.005 *
Days of exercise per week	5 ± 2	5 ± 2	4 ± 2	0.033 *
Walking	37 (57.8%)	28 (66.7%)	9 (40.9%)	0.043 *
Activity at work	13 (20.3%)	10 (23.8%)	3 (13.6%)	0.268 *
Resistance training	6 (9.3%)	5 (11.9%)	1 (4.5%)	0.320 *
Aerobic exercise	4 (6.2%)	4 (9.5%)	0	0.176 *
Cycling	0	0	0	-
TRX	0	0	0	-

Note: The data are presented as frequencies (percentages); * Chi-square Test of independence.

**Table 6 jpm-12-00645-t006:** Changes in body weight before and after hospitalization, views about the effects of hospitalization on the frequency of exercise after discharge and during recovery, effects of the lockdown to the frequency and the desire of the study’s population to exercise and their comparison based on sex, n = 64.

Parameters	Sum n = 64	Men n = 42	Women n = 22	*p*-Value
Changes in body weight after hospitalization				
Weight loss	36 (56.3%)	26 (61.9%)	10 (45.5%)	0.459 *
Weight increase	12 (18.7%)	7 (16.7%)	5 (22.7%)	
Consistent weight	16 (25%)	9 (21.4%)	7 (31.8%)	
Weight change in Kg #	8 ± 6	8 ± 6	7 ± 5	0.388 *
Return to the pre-infection physical condition	56 (87.5%)	39 (92.9%)	17 (77.3%)	0.085 *
Recovery time after hospitalization (days)	35 ± 29	31 ± 28	45 ± 29	0.070 *
Hospitalization effects on the frequency of exercise				
Frequency increase	17 (26.6%)	11 (26.2%)	6 (27.3%)	0.396 *
Frequency decrease	9 (14%)	5 (11.9%)	4 (18.2%)	0.396 *
Consistent frequency	23 (35.9%)	21 (50%)	2 (9%)	0.002 **
No exercise	15 (23.4%)	5 (11.9%)	10 (45.5%)	0.002 **
Effects of exercise on the recovery time				
Negative	2 (3.1%)	1 (2.4%)	1 (4.5%)	0.876 *
Positive	54 (84.4%)	36 (66.7%)	18 (33.3%)	0.703 *
Do Not Know	8 (12.5%)	5 (11.9%)	3 (13.6%)	0.876 *
Lockdown effects on the frequency of exercise				
Frequency increase	17 (26.6%)	11 (26.2%)	6 (27.3%)	0.066 *
Frequency decrease	9 (14%)	5 (11.9%)	4 (18.2%)	0.066 *
Consistent frequency	23 (35.9%)	21 (50%)	2 (%)	0.002 **
No exercise	15 (23.4%)	5 (11.9%)	10 (45.5%)	0.002 **
Lockdown effects on the desire for exercise				
Negative	23 (35.9%)	10 (23.8%)	13 (59%)	<0.001 **
Positive	25 (39%)	16 (38.1%)	9 (40.9%)	0.650 *
Do Not Know	26 (25%)	16 (38.1%)	0	<0.001 **

Note: The data are presented as frequencies (percentages) or mean values ± SD; * Chi-square Test of independence; ** Bonferonni method; # Weight change refers to weight gain or loss.

**Table 7 jpm-12-00645-t007:** Multiple linear regression model to predict a faster recovery time (a).

	Unstandardized Coefficients		Standardized Coefficients	t	Sig.	95.0% Confidence Interval for B	
Model	B	Std. Error	Beta			Lower Bound	Upper Bound
(Constant)	35.021	10.073		3.477	0.007	12,233	57,808
b. Weight loss	2.906	0.225	1.002	9.322	<0.001	1587	2604
b. Chronic Disease	−24.648	5.002	−0.479	−4.927	<0.001	−35,964	−13,331
b. Dyspnea	13.669 Adjusted	3.112	0.485	4.392	0.002	6628	20,710

a. Dependent Variable: recovery time in days; b. Predictors: (Constant), weight loss, chronic disease, dyspnea.

## Data Availability

The data that support the findings of this study are available on request from the corresponding author, OSK.

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
