# Peer review of "Exercise Preferences and Benefits in Patients Hospitalized with COVID-19"

_jpm, 2022, doi:10.3390/jpm12040645_

Round 1
Reviewer 1 Report
N was low, obesity not significant ...should call it mild obesity, and then you found those who were regular exercisers were the obese ones. Not sure there is a lot of merit to this study. Would expect greater number and would expect you would want to report on more significant weight issues. I personally do not see value in this research but there are no major flaws
Author Response
RESPONSE TO REVIEWER 1:
- N was low, obesity not significant ...should call it mild obesity, and then you found those who were regular exercisers were the obese ones. Not sure there is a lot of merit to this study. Would expect greater number and would expect you would want to report on more significant weight issues. I personally do not see value in this research but there are no major flaws
RESPONSE: We are delighted to hear that there are no major flaws in the study. We do recognize that our study was obviously limited by the small sample size. Even though we aimed to have a larger sample size, the actual response rate was much lower. We now acknowledge this realty as a major limitation of our work [page 15, line 531]. This study is the first that evaluated the frequency and type of physical activity among adults previously hospitalized due to COVID-19 in Greece and found that pre-hospitalization physical activity had long-term health benefits in COVID-19 patients.
We are grateful for the time and energy you expended on our behalf.
Reviewer 2 Report
Thank you for the author's contribution to this manuscript. Please see my detailed comments below.
Introduction
- The study highlights the importance of physical activity during the COVID-19 pandemic. However, the introduction missing the empirical evidence, hence I recommend to the authors to expand the introduction with the literature review on PA and COVID-19. It would be interesting to see how PA was decreased during this time. Another aspect that would be interesting is what kind of regulation was used in Greece during the closers and how its effected PA, since some countries the people could leave their home to exercise alone in parks or streets.
- There is another thing that needs to clarify. It seems that the Authors only involve PA as a predictor for obesity, but there are other factors as well (e.g., eating behavior).
Methods
- The methods should be introduced in more detail since there is missing information that makes results hard to understand.
- Please add a procedure and participants subchapter to the methods.
- The link is not working, please delete it!
- I recommend restructuring the "study tools" subchapter. There is only few information regarding the measures (e.g., answer categories). Please add examples and move them into different subchapters.
- 2. subchapter should be under the "study tools" subchapter since it's part of the measures. Furthermore, add more details on it.
- Add reliability measures if possible
Results
- The results are well introduced, but it will increase the quality of the methods include more information.
- The tables are not identical and there is no title for the last table.
- How was frequency of exercise determined as "increase" and "decrease" (Table 6)?
Discussion
- The strong point of this study is the discussion. However, I recommend to the Authors to use more literature, which would help to see the results in an international context.
Author Response
RESPONSE TO REVIEWER 2:
Thank you for the author's contribution to this manuscript. Please see my detailed comments below.
Introduction
- The study highlights the importance of physical activity during the COVID-19 pandemic. However, the introduction missing the empirical evidence, hence I recommend to the authors to expand the introduction with the literature review on PA and COVID-19. It would be interesting to see how PA was decreased during this time. Another aspect that would be interesting is what kind of regulation was used in Greece during the closers and how its effected PA, since some countries the people could leave their home to exercise alone in parks or streets.
RESPONSE: We are appreciative of the support. In the revised manuscript, we have expanded the introduction as suggested (pages 1-2, lines 22-88).
- There is another thing that needs to clarify. It seems that the Authors only involve PA as a predictor for obesity, but there are other factors as well (e.g., eating behavior).
RESPONSE: Thank you for this comment. We totally agree that PA is not the only predictor for obesity but there are other factors as well. We recognize this issue as a limitation of our study and acknowledge it on page 15, line 531.
Methods
- The methods should be introduced in more detail since there is missing information that makes results hard to understand. Please add a procedure and participants subchapter to the methods. The link is not working, please delete it! I recommend restructuring the "study tools" subchapter. There is only few information regarding the measures (e.g., answer categories). Please add examples and move them into different subchapters. 2. subchapter should be under the "study tools" subchapter since it's part of the measures. Furthermore, add more details on it. Add reliability measures if possible
RESPONSE: Thank you for the comment. We have restructured and better defined the Methods section, per your suggestion. We do recognize that reliability measures were not used in the study, but only self-report questionnaires were used to collect data. We now acknowledge this realty as a major limitation of our work (page 15, line 530).
Results
- The results are well introduced, but it will increase the quality of the methods include more information. The tables are not identical and there is no title for the last table.
RESPONSE: We are appreciative of this comment. In order to present data in a more informative way, we have introduced the new Table 3, which has replaced the previous one, presenting comparisons regarding symptoms on admission between patients with the presence or absence of comorbidities. Moreover, the title for the last table has been added. We apologize for this error.
- How was frequency of exercise determined as "increase" and "decrease" (Table 6)?
RESPONSE: Thank you for the comment. In Table 6, weight change refers either to weight gain or loss. We clarified this issue in the table legend (page 10, line 288).
Discussion
- The strong point of this study is the discussion. However, I recommend to the Authors to use more literature, which would help to see the results in an international context.
RESPONSE: Thank you for this interesting remark. In the revised manuscript we have enriched the discussion section (pages 12-15, lines 387-451)
We really thank you for taking the time and energy to help us improve this paper. we very much appreciated your encouraging and insightful comments.
Reviewer 3 Report
Would typically use the term sex not gender.
Table 3 looks at the symptoms present at hospitalization comparing across sex. I question whether this is the most informative way of presenting the data. Is it a sex comparison you are interested in? For instance you could look at comorbidities
Do you have more information about their exercise? Just asking about the number of days they do an activity is not all that informative. Do you have some form of reliability or validity measure for this assessment?
Line 165-167 sounds out of place here if you did not intend to assess this.
Table 6 weight change in kg is this positive or negative?
In addition to improving functions related to cardi-293 ovascular and respiratory function as well as avoiding the deposition of body fat, phys-294 ical activity has many benefits for avoiding infectious diseases, or in the case of hospi-295 talization, for faster recovery [8,16]
I would rephrase the above to something along the lines of… focusing on how physical activity improves inflammation associated with COVID-19 independent of body fat which can explain why body weight is not an independent contributor to hospitalization.
I think you need to balance out the statement below. Given that your age group is over 60 I think there are other factors that could explain the difference between sexes in physical activity engagement.
The factor that affected their pursuit of physical activity differs 300 between the two genders, and the leading cause of this phenomenon may be that women 301 tend to spend more time handling family matters and everyday family needs [17].
There is quite a bit in the discussion about how walking was the best indicator of positive recovery post hospitalization. It is likely that this isn’t the walking itself in regards to intensity and how it affects the body but the fact that you statistically had a high number of individuals in that category and they potentially did a greater volume of walking ( a variable that I don’t believe you collected).
Author Response
RESPONSE TO REVIEWER 3:
- Would typically use the term sex not gender.
RESPONSE: Thank you for the comment. We have used the term sex to replace the term gender throughout the text.
- Table 3 looks at the symptoms present at hospitalization comparing across sex. I question whether this is the most informative way of presenting the data. Is it a sex comparison you are interested in? For instance you could look at comorbidities
RESPONSE: We are appreciative of this comment. Per your suggestion, we have introduced the new Table 3, which has replaced the previous one, presenting comparisons regarding symptoms on admission between patients with the presence or absence of comorbidities.
- Do you have more information about their exercise? Just asking about the number of days they do an activity is not all that informative. Do you have some form of reliability or validity measure for this assessment?
RESPONSE: Thank you for this comment. Unfortunately, in this study self-report questionnaires were used to collect data. We now acknowledge this realty as a major limitation of our work (page 15, lines 525-526).
- Line 165-167 sounds out of place here if you did not intend to assess this.
RESPONSE: Thank you for this comment. The sentence has been removed, as suggested.
- Table 6 weight change in kg is this positive or negative?
RESPONSE: Thank you for the comment. In Table 6, weight change refers to weight gain or loss. We clarified this issue in the table legend (page 10, line 288).
- In addition to improving functions related to cardiovascular and respiratory function as well as avoiding the deposition of body fat, physical activity has many benefits for avoiding infectious diseases, or in the case of hospitalization, for faster recovery [8,16]. I would rephrase the above to something along the lines of… focusing on how physical activity improves inflammation associated with COVID-19 independent of body fat which can explain why body weight is not an independent contributor to hospitalization.
RESPONSE: Thank you for this direction. In the revised manuscript, we have focused on how physical activity improves inflammation associated with COVID-19 independent of body fat which can explain why body weight is not an independent contributor to hospitalization (pages 12-13, lines 387-418), as suggested.
- I think you need to balance out the statement below. Given that your age group is over 60 I think there are other factors that could explain the difference between sexes in physical activity engagement. The factor that affected their pursuit of physical activity differs between the two genders, and the leading cause of this phenomenon may be that women tend to spend more time handling family matters and everyday family needs.
RESPONSE: Thank you for the comment. In the revised manuscript we have enriched the discussion section by reporting several factors contributing to different physical activity between the two genders and comparing our results with research by others (page 13, lines 424-440).
- There is quite a bit in the discussion about how walking was the best indicator of positive recovery post hospitalization. It is likely that this isn’t the walking itself in regards to intensity and how it affects the body but the fact that you statistically had a high number of individuals in that category and they potentially did a greater volume of walking (a variable that I don’t believe you collected).
RESPONSE: Thank you for the comment. In the revised manuscript, we discuss this issue on page 14, lines 445-451.
We appreciate all of your insightful comments. We found them quite useful as we approached our revision. We are grateful for the time and energy you expended on our behalf.
Round 2
Reviewer 2 Report
Thank you for the Authors contribution. All changes have been made. I recommend publishing.